# Enhancing Machine Learning System Reliability in Healthcare through Uncertainty Estimation and Multi-Modal Learning

## Abstract

It is crucial to ensure the dependability of machine learning (ML) systems, especially in areas where safety is a top priority, like healthcare. A tried-and-true method for highlighting the reliability of ML systems during deployment is uncertainty estimation. By successfully using integrated feature sets, sequential and parallel ensemble algorithms have both shown improved ML system performance in multi-modal contexts. We provide Uncertainty-Receptive fusing (URF), a cutting-edge technique that uses uncertainty estimations to improve the fusing of predictions from several base learners. URF, which successively modifies the weighting of the loss function during training in contrast to conventional boosting techniques, is especially successful for multi-modal learning tasks. In order to understand how noise and spatial transformations affect image-based activities, we then offer an image acquisition model that takes these aspects into consideration. We can make predictions with greater accuracy utilizing latent variables thanks to this approach. To quantify uncertainty at the pixel and structure/lesion levels, we use entropy-based uncertainty assessment (EUA). EUA measures the variety within prediction distributions and provides insightful information about the model's confidence. We also present Gnostic Uncertainty Estimation (GUE), which quantifies the model's lack of knowledge regarding the result and helps to comprehend the accuracy of the prediction.

## 1 Introduction

Significant developments in the field of Deep Learning have been made over the past few years, resulting in outstanding improvements in the use of computer vision approaches for the interpretation of medical images Papachristou & Bosanquet (2020); Gal & Ghahramani (2016). To address a variety of disorders, several algorithms have been painstakingly designed and improved Ayhan & Berens (2022). However, the occurrence of batch effects Saad et al. (2010) is a significant obstacle in the field of biomedical image analysis. These effects cover the variances caused by technical artifacts in various data subsets. Differences in sample handling and data collecting procedures complicate matters and provide barriers to the straightforward application of computer vision algorithms to datasets gathered from distinct pathology laboratories. This problem is a major roadblock to the development of machine learning models in this area. Therefore, resolving the complexities of batch effects becomes crucial for the effective creation of precise and reliable ML models specifically designed for medical image analysis.

In situations requiring urgent care in the emergency room (ER), fractures frequently manifest themselves Chen et al. (2023). Bone fractures caused by accidents or diseases like osteoporosis have the potential to cause serious long-term effects or even death. The most effective diagnostic technique for finding bone fractures is to use X-ray imaging of the affected area Factor et al. (2023); Gompels et al. (2023). In emergency departments (EDs), where patients commonly feel discomfort and fractures may not be immediately obvious, this task is especially challenging Lindsey et al. (2018). Healthcare professionals have access to a variety of imaging methods, including X-rays, computed tomography (CT), and magnetic resonance imaging (MRI), for the examination of the musculoskeletal system. Musculoskeletal X-rays stand out as the preferred method for fracture diagnosis among these alternatives. Collaboration between trained radiologists with expertise in mus-

culoskeletal imaging and emergency department doctors in charge of cases involving acute injuries is required for this process. However, difficulties exist in obtaining accurate X-ray interpretations in the emergency room setting, which may cause novice emergency physicians to make unintended mistakes or misclassifications Mall et al. (2023); Jones et al. (2020). In response to this difficulty, image-classification software has risen as a valuable asset in aiding emergency professionals in detecting fractures Karanam et al. (2023). This becomes notably important within emergency rooms, where securing a second opinion frequently proves unfeasible. Through the utilization of such software, medical providers can elevate their diagnostic precision, leading to an overall enhancement in patient care during critical situations.

Over 1.7 billion people worldwide are affected by musculoskeletal problems, which are the main cause of excruciating, ongoing suffering as well as impairment. These problems are getting worse as seen by the rising number of 30 million visits to emergency rooms annually. We hope that our MURA-dataset Rajpurkar et al. (2017) will open the door to significant advancements in the field of medical imaging technologies. These developments have the potential to lead to expert-level diagnoses, ultimately improving access to healthcare in areas where the number of skilled radiologists is still scarce. The necessity for automated fracture detection is paramount in curbing the advancement of acute injuries through timely patient diagnoses. Traditional methods involving radiologists often entail substantial resource allocation. Consequently, the inclination to utilize deep neural networks for the automatic categorization of fractures in X-rays has experienced substantial growth in recent years Kandel & Castelli (2021); He et al. (2021).

A well-tuned classifier would give ambiguous categories a lower likelihood. In the medical area, where it's critical to have faith in the model's reliable predictions for screening automation and also guide cases of doubt toward manual evaluation by medical personnel, the issue of assessing uncertainty is particularly important Rahaman et al. (2021). Making machine learning models with a clear and inbuilt knowledge of uncertainty is made possible by the Bayesian probability theory Yang & Fevens (2021). Such models are capable of calculating the mean and variance of the output distribution for each class, in addition to other parameters, rather than just a single per-class probability.

Numerous probabilistic and Bayesian procedures, as shown by Graves (2011); Hernández-Lobato & Adams (2015); Pearce et al. (2020); Yang et al. (2019); Wu et al. (2018), as well as alternative non-Bayesian strategies like Sinha et al. (2019); Zhang et al. (2021); Nomura et al. (2021); Dusenberry et al. (2020), have been presented to efficiently assess uncertainty estimates. These techniques are essential for measuring how trustworthy and confident Neural Networks (NNs) are in their predictions. Additionally, the use of network ensembles has become a tactic to improve the general performance of models. The idea of ensemble methods has advanced significantly in the goal of incorporating uncertainty estimates thanks to methods like those covered in Kendall et al. (2018). These methods combine the method of training numerous models to handle various tasks with the use of anticipated uncertainties as weights for the individual model losses. This novel approach not only outperforms the performance of specialized models trained for each task, but also offers a more reliable and well-informed decision-making process. While non-Bayesian alternatives provide an easy and practical method for estimating uncertainty in the context of deep neural networks, the ensemble approach - while conceptually basic presents a real-world problem due to its resource-intensive nature. This becomes especially clear when many networks are used simultaneously during the training and inference phases.

We provide an end-to-end system specifically created for the classification of fractures musculoskeletal radiographs images, offering a novel and creative method. This framework combines the ideas of EUAug Wang et al. (2019) and Uncertainty-Receptive Fusion (URF), drawing influence from both, to create a coherent and effective strategy. Thus, our methodology makes it easier to produce final forecasts that are accurate and well-calibrated. Our method's ability to smoothly include an ensemble of predictions while taking the model's inherent uncertainty into account is one of its distinctive characteristics. This integration guarantees the accuracy and dependability of the results while maintaining the fundamental efficiency of the model. In essence, our method finds a balance between accuracy and robustness, producing outcomes that inspire confidence in categorization decisions for fractures musculoskeletal radiographs images.

## 2 METHODOLOGY

The proposed methodology comprises two subparts (1) for uncertainty estimation, (2) discusses in-depth the acquisition of images, the evaluation of uncertainty (including entropy-based and gnostic methods), and the calculation of structural uncertainty using the Volume Variation Coefficient (VVC). The following fusion methods were created for multi-modal regression projects. They tackle the problem of mixing data from many input modalities, each linked to a different base learner, to provide precise predictions. While the modified version adds weighted aggregation based on inverse uncertainty measures, it concentrates on including uncertainty estimations. The image acquisition methodology, uncertainty assessment techniques (Entropy-based Uncertainty Assessment and Gnostic Uncertainty estimate), and structural uncertainty estimate utilizing the Volume Variation Coefficient (VVC) are all covered in detail in the second part.

### 2.1 UNCERTAINTY-RECEPTIVE FUSION

For the purposes of regression, let's assume that $I_m$ represents the input feature set with many modes and $y \in \mathbb{R}$ represents a real-valued label. The collection of $d$ dimensional input properties directly connected to the $j^{th}$j modality is represented by the set $I_j \in \mathbb{R}^d$. Here, $j$ is an integer between 1 and $m$, with $m$ denoting the total number of modalities. Each modality's matching base learner, represented as $\{h_j\}_{j=1}^m$, serves as a representation for the learned functions that can map inputs to outputs. As the underlying learning process, these functions may use a variety of methods, including SVMs, KNN, decision tree, random forests, neural networks, and comparable models. Our training dataset is then made up of $N$ independent and identically distributed (i.i.d.) samples labeled as $\{(I_{jn}, y_n)\}_{n=1}^N$, precisely corresponding to the $j^{th}$ modality.

We start by constructing a baseline fusion procedure, which we refer to as the Vanilla Fusion (VF), to ensure a fair assessment. Then, we describe our invention, known as the Uncertainty-Receptive Fusion (URF), and its modified form, known as URF$_w$. 1 shows a graphic illustration of the VF, URF, and URF$_w$ methods.

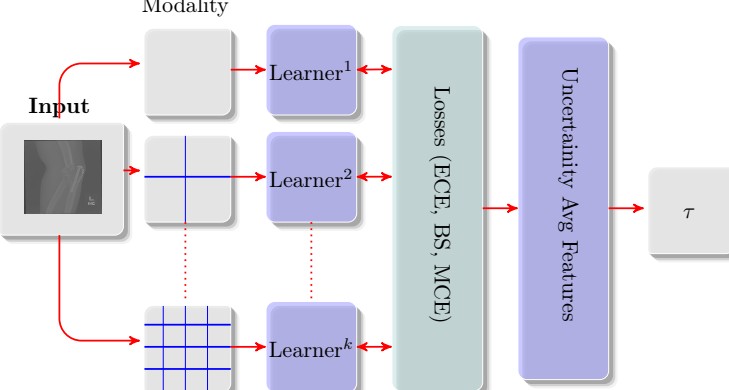

Figure 1. Our approach uses the Mura image dataset, which consists of a variety of musculoskeletal radiograph images, each representing a different class, such as "normal", "fracture", and "arthritis". To increase the dataset's diversity and improve model generalization, we apply data augmentation techniques along with patching, which involves dividing images into smaller sections. We also provide a method to average and capture the uncertainty in these output characteristics. In the domain of medical image analysis, this is crucial since it enables us to take into account prediction uncertainty and produce more accurate findings for clinical applications.

In Vanilla Fusion (VF), the weighting of the loss function is adjusted during training while iteratively boosting among the base learners using loss data, such as Mean Squared Error (MSE) for regression, Kullback-Leibler (KL) divergence loss the discrepancy between the actual distribution of class probabilities and the expected class probabilities and IoU or Jaccard Loss evaluates the overlap between anticipated and ground truth masks and is frequently used in semantic segmentation problems. The error values connected to the predictions made by the $j^{th}$ base learner are specifically used to adjust

the weight of the loss function for the related training instances during the training of the $(j + 1)^{th}$ base learner. The ensemble then calculates the average of all the predictions made by the boosted base learners $\{\hat{y}_{h_j}\}_{j=1}^{m}$ to arrive at the final prediction $\hat{y}$.

In Uncertainty-Receptive Fusion (URF), the method successively boosts through the base learners while adjusting the weighting of the loss function during training using the predicted uncertainty estimations $\sigma_{h_j}$. In practice, this requires adjusting the weight of the loss function with regard to the pertinent training instances during the training of the $(j + 1)^{th}$ base learner using the uncertainty estimates $\sigma_{h_j}$ associated with predictions made by the $j^{th}$ base learner. The ensemble then determines the average of all the predictions $\{\hat{y}_{h_j}\}_{j=1}^{m}$ derived from all the boosted base learners to get the final prediction $\hat{y}$.

We investigate a change to the previously mentioned UA ensemble's final prediction aggregation. In this variation, the ensemble computes a weighted average of the predictions $\{\hat{y}_{h_j}\}_{j=1}^{m}$ from all the boosted base learners for the final prediction $\hat{y}$. The inverses of the corresponding anticipated uncertainty estimations, or $\sigma_{h_j}$ are used to calculate these weights. 2 mathematically illustrates this, where $m$ is the overall number of distinct modalities and $\hat{y}(i_n)$ is the final forecast for the $n^{th}$ data point.

$$\hat{y}(i_n) = \frac{\sum_{j=1}^{m} \sigma_{hj}(i_n)\hat{y}_{hj}(i_n)}{\sum_{j=1}^{m} \sigma_{hj}(i_n)} \tag{1}$$

In this context, $I_n$ denotes the $n^{th}$ input images, and $\hat{y}_{hj}(i_n)$ denotes the $n^{th}$ input image's output for the $j^{th}$ model's prediction. The predictions from the $j^{th}$ model correspond to the uncertainty estimate, given as $\sigma_{hj}$. The independent uncertainty metric's inverse is calculated to create the uncertainty weights Sarawgi et al. (2021). As a result, $\hat{y}(i_n)$ produces the final forecast for the $n^{th}$ data point.

In this situation, a modified version of LLFU Lakara et al. (2021) is used to measure the uncertainty associated with each prediction:

$$
\begin{aligned}
\alpha &= \left( \frac{(y_j(i_n) - \mu(i_n))^2}{2\sigma^2(i_n)} \right) \\
\beta &= \max\left( 0, \log\left( \frac{2\pi\sigma^2(i_n)}{2} \right) \right) \\
\gamma &= \left( \frac{1}{2\sigma^2(i_n)} \right) \\
\sigma_{hj} &= \sqrt{\alpha + \beta + \gamma}
\end{aligned}
\tag{2}
$$

In this equation, $y_j(i_n)$ stands for the prediction associated with the $j^{th}$ model, $\mu(i_n)$ stands for the mode of predictions from all the models in the ensemble, and $\sigma^2(i_n)$ stands for the standard deviation of predictions for the $n^{th}$ data point.

URF and URF$_w$ follow a novel pattern of sequential boosting among diverse base learners, in contrast to the majority of prior boosting techniques that incrementally boost using the same collective input characteristics. Each fundamental learner is matched with a particular input modality. By combining the efforts of individual modality-specific base learners, this system attempts to maximize the exploitation of modality-specific features while creating a solid multi-modal learner. It is important to stress that, unlike other boosting strategies Chen & Guestrin (2016), the base learners in this case are not just weak learners.

## 2.2 Techniques for Acquiring Images and Their Transformations

The method used to acquire the observed images is described in the model for obtaining images. Numerous factors, whether linked to or unrelated to the imaged topic, have an impact on this process. These parameters include elements like down- and up-sampling, blue, spatial transformation, and system noise. Though down- and up-sampling and blurring are frequently discussed in relation

to image super-resolution, their impact on image identification is very negligible. It is possible to smoothly combine complicated intensity variations or other data augmentation methods like elastic deformations. As a result, we concentrate on spatial change and noise. The following is the image acquisition model:

$$M = \mathbb{O}\left(\mathbb{P}(I_o)\right) + n \tag{3}$$

In this case, $I_0$ represents a latent image with a particular position and orientation, which is essentially a hidden variable. The transformation operator used on $I_0$ is represented by the symbol $\mathbb{P}$. The injected noise in the converted image is represented by $n$, and the parameters of this transformation are marked by the symbol. The observed image, which is used to draw conclusions when testing, is designated as $I$. Although changes in space, intensity, or feature space might also be considered transformations, this study only looks at the impact of reversible changes in space, such as flipping, scaling, rotation, and translation. These alterations are used to supplement data and are most frequently used during image capture. We obtain the following expression by designating $\mathbb{O}(\mathbb{P})$ inverse transformation as $\mathbb{O}(\mathbb{P})^{-1}$:

$$M_0 = \mathbb{O}(\mathbb{P}^{-1}(I - n)) \tag{4}$$

We operate under the presumption that the distribution of $I$ includes the distribution of $I_0$, similar to the augmentation of data during training. This presumption results in specific prior distributions for the noise and transformation parameters in a particular setting. Consider a 2D image of musculoskeletal radiographs as an illustration, where the orientation of the musculoskeletal radiographs can cover all conceivable directions inside the 2D plane. Consequently, a homogeneous prior distribution $r \sim U(0, 2\pi)$ can be used to define the rotation angle $r$. When modeling image noise, the Gaussian distribution is frequently used, i.e., $n \sim N(\mu, \sigma)$, where $\mu$ and $\sigma$ stand for mean and standard deviation, respectively. We therefore have $\mathbb{P} \sim p(\mathbb{P})$ and $n \sim p(n)$, denoting the previous distribution of $\mathbb{P}$ as $p(\mathbb{P})$ and that of $n \sim p(n)$.

Consider $Y$ and $Y_0$ as labels corresponding to $X$ and $X_0$, respectively. $Y$ and $Y_0$ serve as categorical variables in the context of image classification, which makes it necessary for them to be invariant to noise and changes, therefore, $Y = Y_0$. In image segmentation, the discretized label maps $Y$ and $Y_0$ have behavior that is consistent with the spatial transformation $Y = \mathbb{O}(\mathbb{P}Y_0)$.

### 2.2.1 PRIOR DISTRIBUTIONS OF FLUKY MODEL

Consider $f(\cdot)$ to be the function embodied by a neural network in the circumstance of deep learning, and let $\theta$ signify the parameters obtained from a sample of learning images together with their related annotations. In the standard configuration, the anticipated $Y$ of a test image $I$ is determined by:

$$Y = f(\theta, X) \tag{5}$$

The term $Y$ is used in relation to continuous values in regression tasks. $Y$ represents discretized labels obtained using an argmax operation within the network's top layer for segmentation or classification problems. As $I$ is only one possible observation of the underlying image $I_0$, direct inference using $I$ may result in biased results that are affected by the particular mathematical function and noise related with $I$. To address this issue, our goal is to draw conclusions using the latent variable $I_0$ instead:

$$Y = \mathbb{O}(\mathbb{P}(Y_0)) = \mathbb{O}(\mathbb{P}f(\theta, X_0)) = \mathbb{O}(\mathbb{P}f(\theta, \mathbb{O}^{-1}(\mathbb{P}((I - n))))) \tag{6}$$

Since the specific values of $\mathbb{P}$ and $n$ for $I$ are still unknown, we instead concentrate on the distribution of $Y$ to provide robust inference. We also take into account the distributions of $\mathbb{P}$ and $n$ for $I$.

$$p(Y|I) = p\big(\mathbb{O}\big(\mathbb{P}f\big(\theta, \mathbb{O}\big(\mathbb{P}^{-1}(I - n)\big)\big)\big)\big) \tag{7}$$

where $\mathbb{P} \sim p(\mathbb{P})$ and $n \sim p(n)$. For regression tasks, we compute the mean value of $Y$ using the statistical distribution $p(Y|X)$ and then use that expectation to determine the final prediction for $I$.

$$
\begin{aligned}
E(Y|I) &= \int y \cdot p(y|I) \\
Q &= p(\mathbb{P})p(n)\, d\mathbb{P}\, dn \\
dy &= \int_{\mathbb{P} \sim p(\mathbb{P}), n \sim p(n)} \mathbb{O}\left(\mathbb{P}\left(f(\theta, \mathbb{O}(\mathbb{P}^{-1}(I - n))) \cdot Q\right)\right)
\end{aligned}
\tag{8}
$$

Given the continuous nature of $\mathbb{P}$ and $n$ and the sophisticated joint distribution $p()$ including several transformation types, computing $E(Y|I)$ using 8 can be resource-intensive. As an alternative, we use Monte Carlo simulation to approximate $E(Y|I)$. In this case, $N$ stands for the total number of simulation iterations. The prediction is made in the nth simulation iteration as follows:

$$y_j = \mathbb{O}(\mathbb{P}_j f(\theta, \mathbb{O}(\mathbb{P}_n^{-1}(I - n_j)))) \tag{9}$$

In this example, $n_j$ is sampled from $p(n)$ while $\mathbb{P}_j$ is taken from the distribution $p(\mathbb{P})$. We begin the process by selecting $\mathbb{P}_j$ and $n_j$ at random from the prior distributions $p(\mathbb{P})$ and $p(n)$, respectively, in order to get $y_j$. Then, using $\mathbb{P}_j$ and $n_j$, we build a probable hidden image as described in equation 4. The trained network then receives this image as input to make a prediction, which is then further altered with $\mathbb{P}_j$ to produce $y_j$ as specified in Equation 6. The set $Y = y_1, y_2, ..., y_J$ assembled from the distribution $p(Y|I)$ allows us to calculate $E(Y|I)$ by averaging $Y$, and this average serves as our final prediction $\hat{Y}$ for $I$:

$$\hat{Y} = E(Y|I) \approx \frac{1}{J} \sum_{j=1}^{J} y_j \tag{10}$$

The distribution $p(Y|I)$ often transforms into a discrete distribution for classification or segmentation tasks. Maximum likelihood estimation is used to determine the final forecast for $I$.

$$\hat{Y} = \arg\max p(y|I) \approx Mod\,(Y) \tag{11}$$

The element in $Y$ that appears the most frequently is referred to as $Mod(Y)$ in this context. This idea is compatible with the notion of combining various predictions using a majority voting strategy.

### 2.2.2 Entropy-based Uncertainty Assessment (EUA)

By measuring the range of deviations among predictions provided for a particular image, uncertainty is assessed. Variance and entropy are two metrics that can be used to measure the diversity within the distribution $p(Y|I)$. However, variance is not a complete indicator when dealing with distributions that have several modes. Entropy is the metric of choice for estimating uncertainty in this investigation.

$$H(Y|I) = -\int p(y|I) \ln p(y|I)\, dy \tag{12}$$

We may calculate $H(Y|I)$ using the Monte Carlo (MC) simulation, based on the simulated results $Y = y_1, y_2, \ldots, y_J$ . If $Y$ contains $V$ distinct values, which frequently correspond to labels in classification tasks, and the frequency of occurrence of the $v^{th}$ unique value is indicated by the symbol $\hat{p}_v$, then an approximation of $H(Y|I)$ can be written as follows:

$$H(Y|I) \approx -\sum_{v=1}^{V} \hat{p}_v \ln(\hat{p}_v) \tag{13}$$

It is advantageous to assess uncertainty at the pixel level when doing segmentation tasks. The anticipated label for the $i^{th}$ pixel is shown here as $Y_i$. A set of values for $Y_i$ is obtained by the use of a Monte Carlo simulation, as shown by $Y^i = y_1^i, y_2^i, \ldots, y_J^i$ . As a result, the entropy of the $Y_i$ distribution can be roughly calculated as follows:

$$H(Y^i|I) \approx -\sum_{v=1}^{V} \hat{p}_v^i \ln(\hat{p}_v^i) \tag{14}$$

where $\hat{p}_v^i$ represents the occurrence frequency of the $v^{th}$ unique value in $Y^i$.

### 2.2.3 Gnostic Uncertainty Estimation (GUE)

We use the run-time dropout and Entropy-based Uncertainty Assessment (EUA) method to derive estimates of model (gnostic) uncertainty. In this method, the network parameter set $\theta$ is approximated by $q(\theta)$, where the individual members are stochastically assigned zero values based on Bernoulli random variables. By reducing the Kullback-Leibler (KL) separation between $q(\theta)$ and

the posterior distribution of $\theta$ given a learning dataset, it is possible to realize $q(\theta)$. Following the training phase, the following is expressed as the prognostic organization for a test image $I$:

$$p(Y|I) = \int p(Y|I, \omega) q(\omega) \, d\omega \tag{15}$$

Using the trained network and a technique called MC dropout, Monte Carlo iterations can sample the forecast distribution. The output of the function $f(\theta_j, I)$, where $\theta_j$ is a Monte Carlo sample taken from the distribution $q(\theta)$, yields each sample $y_j$ in the proposed technique. The set of these sampled $Y$ values is $Y = y_1, y_2, \ldots, y_J$ if the total number of samples is $N$. As a result, for regression problems or classification/segmentation tasks, the final prediction for input $I$ can be computed using Equation 10 or Equation 11. The variance or entropy of these $J$ predictions can be calculated to gauge gnostic uncertainty. We apply entropy for this, which corresponds to the idea shown in Equation 14, in order to retain consistency with our EUA methodology. The idea of run-time dropout can be viewed as a technique for building network ensembles for testing. The idea of explicitly using neural network fusions was presented as an alternate method to run-time dropout for measuring gnostic uncertainty.

To calculate uncertainty on a structure/lesion level, the author Nair et al. (2020) used Monte Carlo instances produced via test-time dropout. We broaden the method for structure-wise uncertainty estimate by building on their methods. This extension includes Monte Carlo samples that were acquired using the EUA method as well as test-time dropout. Let's refer the stack of slices of the segmented structure from the $J$ samples generated by the Monte Carlo simulation as $V = v_1, v_2, \ldots, v_J$, where $v_i$ represents the amount of the segmented structure in the $i^{th}$ framework. The symbols $\mu_V$ and $\sigma_V$, respectively, represent for V's mean value and standard deviation. We apply the Volume Variation Coefficient (VVC) as $\mathbb{V}$ to find the structural uncertainty:

$$\mathbb{V} = \frac{\sigma_V}{\mu_V} \tag{16}$$

In this case, the $\mathbb{V}$ maintains its independence from the segmented structure's size.

## 2.3 DATASET

MURA, a sizable database of musculoskeletal radiographs, is now available. It contains 40,561 images derived from 14,863 investigations. Each study in this dataset has been painstakingly classified by radiologists as either normal or pathological. They obtain supplemental comments from six board-certified Stanford radiologists expressly for the exam set for the purposes of thorough review and to determine radiologist proficiency. This collection of 207 musculoskeletal studies serves as a reliable standard for evaluating models and estimating radiologist performance Rajpurkar et al. (2017).

Findings and results are described in Appendix B.

## 2.4 SUMMARY

In contrast to bigger datasets with realistic images like PASCAL VOC, COCO, and ImageNet, the number of training images at our disposal was very small in our trials. The inherent difficulties of curating huge datasets for medical image segmentation give rise to this difference. Radiologists' specialized knowledge is required for the time-consuming task of gathering pixel-by-pixel comments for medical images. As a result, the majority of current medical image segmentation datasets, including those included in Grand Challenge 4, often include a limited number of images.

The assessment of CNNs under these restrictions and with little training data is of utmost relevance to the field of medical image processing. In addition, despite its limited size, our dataset lends itself to data augmentation, supporting our justification for doing so during both the training and testing phases. It's important to note that while working with smaller datasets, the need for uncertainty estimate increases significantly.

Spatial transformations and image noise were explicitly included in our statistical conceptualization for testtime step-up, which was built on an image acquisition technique. This system, however, is easily scalable to consider more thorough mathematical functions, such elastic deformations Çiçek

et al. (2016), or to add simulated bias fields for X-rays. Beyond the variation in model parameter values, elements in the input data such image noise and object-related modifications also have an impact on the prediction result. Consequently, a proper cognitive state calculation should take these factors into cerebration. Figures 6 provide compelling evidence that depending simply on model uncertainty frequently leads to unduly optimistic but inaccurate forecasts. In this situation, EUA shows up as a key element in preventing similar situations. Furthermore, Figure 5 displays five sample instances, with each subfigure providing the outcomes for a different patient, and Table 4 provides in-depth statistical analysis for all testing images. Our results show that EUA + TTD did not outperform EUA in terms of attaining better Dice scores in a few cases involving certain testing images. The average Dice score of EUA + TTD marginally surpassed that of EUA when taking into account the total performance across all testing images, nevertheless. The data in Figures 6 further support our conclusion that EUA + TTD does not always outperform EUA and that, on average, their performance stays closely linked.

We used the setting of image segmentation tasks to explain how EUA may be used. However, it is applicable to a variety of image recognition applications, including object identification, regression, and image classification. Entropy may not be the most appropriate metric for estimating uncertainty in regression tasks, especially when the findings are not discrete class labels. Instead, the output distribution variation may be more appropriate. The benefits of test-time augmentation in improving segmentation accuracy are clearly seen in Table 5. It also emphasizes how well W-Net may be combined from many angles to improve overall performance. As previously shown by Lakshminarayanan et al. (2017), this amalgamation is an aggregation of different networks and may be investigated as a different method for estimating epistemic uncertainty.

In our investigation, we found that the Monte Carlo sample size $N$ that achieves a plateau in segmentation accuracy often falls within the range of 20 to 60 for the CNNs we examined and the particular applications under consideration. Empirically, we discovered that a value of $N$ of 60 is suitable for our datasets, resulting in a sufficiently sized sample. It's important to remember, though, that depending on the dataset, the best hyper-parameter $N$ choice may change. When the ideal $N$ is fewer than 40, using a fixed $N$ number for new applications might cause additional computing overhead and perhaps reduce efficiency. On the other hand, the ideal $N$ value may exceed 40 in applications with more apparent spatial fluctuations inside objects. As a result, when starting a new application, we advise carefully experimenting to find the optimal $N$, ideally by locating the point at which performance stabilizes on a validation set.

## 3 Conclusion

In conclusion, our research examined a variety of aspects of uncertainty in CNN-driven medical image segmentation. We accomplished this by contrasting and combining uncertainty that are model-derived (epistemic) and input-influenced (impromptu). The creation of a test-time augmentation-based technique for calculating impromptu uncertainty in medical images that takes into consideration the effects of both image noise and spatial changes was a significant contribution. In addition, we presented a thorough theoretical and mathematical foundation for test-time extension. This methodology included modeling the previous distributions of parameters inside an image acquisition model and creating a prediction distribution using Monte Carlo simulation. Our research, which included both 2D medical image segmentation tasks, demonstrated the value of our EUA strategy for estimating uncertainty. When depending simply on model-based uncertainty estimates, overconfident inaccurate predictions, which frequently occur, are successfully minimized by EUA. It is noteworthy that EUA regularly produced segmentation accuracy that was greater than both the baseline of a single prediction and even many predictions acquired using test-time dropout.

### Author Contributions

If you'd like to, you may include a section for author contributions as is done in many journals. This is optional and at the discretion of the authors.

ACKNOWLEDGMENTS

Use unnumbered third level headings for the acknowledgments. All acknowledgments, including those to funding agencies, go at the end of the paper.

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

## APPENDICES

## A    MEASUREMENT METRIC

Cohen's quadratic weighted Kappa ($\kappa$) is used to measure the level of agreement between raters in circumstances requiring ordinal multi-class difficulties while evaluating medical imaging performance [31]. This metric emphasizes rating differences, and as shown below, the difference between the forecast and the actual value has a quadratically inverse relationship to the penalty for disparities.

$$\kappa = 1 - \frac{\sum_{i,j}^{N} w_{i,j} D_{i,j}}{\sum_{i,j}^{N} w_{i,j} P_{i,j}} \tag{17}$$

where $\kappa$ represents Cohen's Quadratic Weighted Kappa. $D_{i,j}$ denotes the discovered agreement between raters for category $i$ and $j$. $P_{i,j}$ stands for the predicted agreement between raters for category $i$ and $j$. $w_{i,j}$ is a weight that can be applied to the elements based on the distance between the categories, which we might have defined in our context.

The goal of the above equation 17 is to identify agreement that goes above what would be predicted by chance. To give a normalized measure of inter-rater agreement, it takes into account both the observed agreement and the chance agreement. More agreement among raters than could be obtained by chance is indicated by a higher Kappa value, whereas a lower number suggests less agreement. The Kappa value varies from $-1 to 1$: Perfect agreement is indicated with a value of 1. A number of 0 represents agreement that is only a matter of chance. The idea that agreement is worse than random is implied by a number smaller than 0.

To measure and quantify the level of uncertainty in our analysis, we use three different metrics. These measurements provide insight into how projections and actual results correlate while also taking uncertainty into account. The three metrics we use are as follows:

### A.1 EXPECTED CALIBRATION ERROR (ECE)

This assessment effectively quantifies the alignment of the model's confidence levels by measuring the difference between predicted probability and their actual manifestations. A lower ECE score means that the model's predictions are more accurate and precisely calibrated Nixon et al. (2019).

$$ECE = \sum_{m=1}^{n} \frac{|B_m|}{n} \cdot |clo(B_m) - sur(B_m)| \tag{18}$$

where

$$clo(B_m) = \frac{1}{|B_m|} \sum_{i \in B_m} 1(y_i = y_t) \tag{19}$$

$$sur(B_m) = \frac{1}{|B_m|} \sum_{i \in B_m} p_i \tag{20}$$

where $ECE$ represents the Expected Calibration Error. $n$ is the number of bins. $B_m$ denotes the $m$-th bin. $clo(B_m)$ stands for the closeness of the bin $B_m$. $sur(B_m)$ represents the average surety of the bin $B_m$. $1(y_i = y_t)$ is the indicator function that evaluates to 1 if $y_i$ is equal to $y_t$, and 0 otherwise. The sum is taken over all $i$ values within the bin $B_m$. $p_i$ denotes the confidence value for instance $i$ in bin $B_m$.

### A.2 MAXIMUM CALIBRATION ERROR (MCE)

This statistic measures the largest discrepancy between actual findings and predicted probability Kumar et al. (2018). It helps identify situations where the model's estimate of confidence dramatically differs from actual results, highlighting possible calibration-related areas that need addressing.

$$MCE = \max_m |clo(B_m) - sur(B_m)| \tag{21}$$

The terms $clo(B_m)$ and $sur(B_m)$ in this equation stand for the closeness and average surety of the bins, respectively, while $MCE$ stands for the Maximum Calibration Error. The formula determines the largest possible absolute difference between the closeness and surety values for all bins $B_m$.

### A.3 BRIER SCORE

By comparing the projected probability with the actual results, the Brier Score evaluates the accuracy of probabilistic forecasts. It takes into account the calibration and resolution of the model's forecasts,

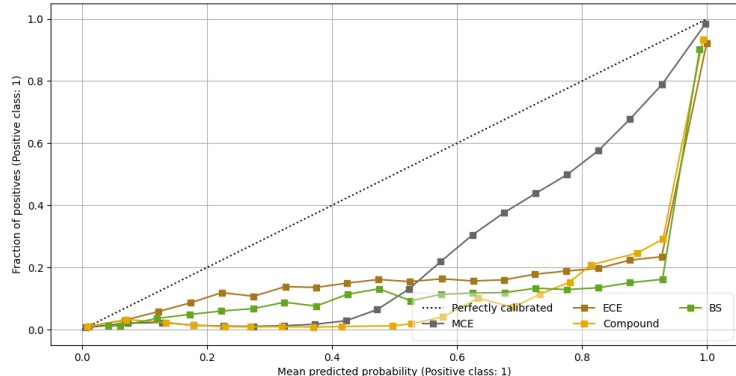

Figure 2. Maximum Calibration Error (MCE), Brier Score (BS), and Expected Calibration Error (ECE) are three measures that are frequently employed for this purpose. These measures make it possible to evaluate how well predicted probability and actual results match up in a model.

with lower numbers signifying better performance Ferro & Fricker (2012).

$$BS = \frac{1}{n} \sum_{t=1}^{n} (f_t - o_t)^2 \qquad (22)$$

where $BS$ represents the Brier Score. $n$ is the number of instances. $f_t$ represents the forecasted probability for instance $t$. $o_t$ represents the observed outcome (actual result) for instance $t$. The sum is taken over all instances from $t = 1$ to $n$.

## B  FINDING AND DISCUSSION

We do several iterations of both training and testing evaluations in order to increase the dependability of our results, and we then compute the mean and variance of the Root Mean Squared Error (RMSE) values over a range of epochs. Prior to comparing each modality (base learner) to the VF and URF models, we first evaluate each modality's performance on its own. Based on the test sample results of the distinct modalities, we decide in what order to enhance these models to propagate uncertainty. Notably, according to our findings, the URF performs better than both the VF model and each of the individual modalities (see Table 1 for visual nformation see Figure 2). In order to evaluate a model's capacity to deliver precise probability estimates for its predictions, confidence calibration is an essential component of machine learning. Maximum Calibration Error (MCE), Brier Score (BS), and Expected Calibration Error (ECE) are three often used metrics to assess confidence calibration. These measures enable us to evaluate the consistency between a model's projected probability and the actual results. Probability calibration curves are visual representations of these metrics that offer insightful information about the calibration effectiveness of a model.

Table 1. Comparison of RMSE Values for Different Models

| SOTA Vs Proposed | RMSE | MCE | BS | ECE |
|---|---|---|---|---|
| Pappagari et al. (2020) | 5.37 | 0.59 | 0.31 | 0.29 |
| Sarawgi et al. (2020) | 4.60 | 0.61 | 0.32 | 0.31 |
| Rohanian et al. (2021) | 4.54 | 0.57 | – | 0.29 |
| VF | 3.91 | 0.30 | 0.17 | 0.19 |
| URF | 3.87 | 0.27 | 0.16 | 0.18 |
| URF$_w$ | 3.53 | 0.25 | 0.14 | 0.14 |

Additionally, the Mean Prediction Interval Width (MPIW) and Prediction Interval Coverage Probability (PICP), two extensively used metrics for evaluating the level of uncertainty in regression, are used to examine our technique. While MPIW evaluates the average width or breadth of all prediction intervals, PICP estimates the proportion of cases in which the prediction interval effectively

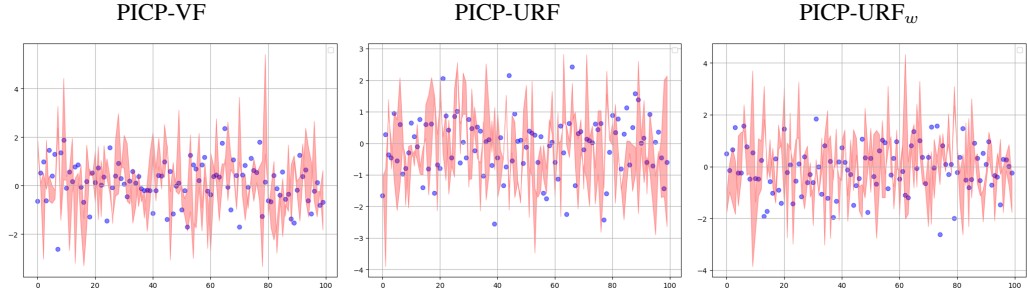

Figure 3. The performance of these prediction intervals is assessed using PICP. It calculates the proportion of times the genuine outcome, or ground truth, is actually present in the prediction interval for medical imaging. The prediction intervals are more trustworthy and effectively represent the uncertainty when the PICP value is greater.

contains the real regression value. Notably, it is stressed in the work by Pearce et al. (2018) that optimal high-quality prediction intervals should be tight while yet catching a specific percentage of the data points. As a result, lowering MPIW values is preferable, and PICP values larger than or equal to (1-$\alpha$), which normally has a common value of 0.05, are also favored. A prediction interval

Table 2. Performance Metrics for Different Models and Modalities

| Model | Modality | MPIW | PICP | | |
| --- | --- | --- | --- | --- | --- |
| | | | $\Delta = 1\sigma$ | $\Delta = 2\sigma$ | $\Delta = 3\sigma$ |
| VF | Elbow | $4.51 \pm 0.51$ | $77.89 \pm 2.05$ | $91.36 \pm 2.64$ | $92.35 \pm 4.61$ |
| | Shoulder | $5.11 \pm 0.94$ | $91.65 \pm 2.67$ | $92.37 \pm 2.64$ | $94.36 \pm 3.81$ |
| | Forearm | $4.16 \pm 1.08$ | $94.36 \pm 3.25$ | $93.27 \pm 2.71$ | $92.38 \pm 2.56$ |
| | Humerus | $5.73 \pm 0.94$ | $94.69 \pm 2.51$ | $95.05 \pm 3.64$ | $94.68 \pm 4.06$ |
| URF | Elbow | $4.11 \pm 1.01$ | $5.46 \pm 1.57$ | $95.68 \pm 2.05$ | $95.78 \pm 2.86$ |
| | Shoulder | $4.05 \pm 0.93$ | $96.01 \pm 3.65$ | $96.43 \pm 3.51$ | $96.79 \pm 2.06$ |
| | Forearm | $4.16 \pm 0.41$ | $96.89 \pm 1.17$ | $97.01 \pm 1.06$ | $98.06 \pm 2.01$ |
| | Humerus | $4.21 \pm 0.35$ | $97.16 \pm 2.11$ | $98.16 \pm 1.69$ | $96.12 \pm 1.74$ |
| URF$_w$ | Elbow | $3.73 \pm 0.56$ | $98.12 \pm 0.95$ | $97.36 \pm 0.97$ | $93.51 \pm 2.17$ |
| | Shoulder | $3.70 \pm 0.47$ | $97.82 \pm 0.65$ | $96.71 \pm 2.41$ | $97.68 \pm 0.26$ |
| | Forearm | $3.90 \pm 0.76$ | $97.83 \pm 1.96$ | $96.79 \pm 0.64$ | $98.34 \pm 0.58$ |
| | Humerus | $3.64 \pm 0.98$ | $97.36 \pm 0.79$ | $96.78 \pm 0.49$ | $98.14 \pm 0.58$ |

is a statistical range or interval that is generated around a point estimate (often the anticipated mean or expected value) of a variable in the context of prediction interval coverage probability (PICP). It helps to measure the degree of uncertainty or variability surrounding a prediction in regression analysis. Figure 3 depicts the interval of the proposed methodology.

The unique model $V - NET^{++}$ compared with identical U-Net and V-NET model that was trained utilizing data augmentation produced the results. In order to evaluate EUA, GUE, and a combination of our proposed uncertainty methodologies, three distinct scenarios – Test-Time Dropout (TTD), EUA augmentation, and TTD + EUA – are each subjected to 20 rounds of Monte Carlo simulations. The classification outcomes and the uncertainty maps for each of these three categories of uncertainties. The pixel-wise entropy over the $N$ predictions is evaluated to produce the uncertainty maps, which are displayed graphically using the color bar in the upper left corner and are found in the odd columns.

As shown by TTD, the majority of the classifications that are debatable are found in the periphery of the foreground that has been classified. Indicating low uncertainty are pixels that are substantially more confident than those that are closer to the boundary. The GUE uncertainty map also has some random noise in the musculoskeletal radiography area. Contrarily, EUA uncertainty computed with EUA-augmentation exhibits less random noise and finds doubtful classification not just along the border but also in difficult regions. The EUA generated result tends to overclassify in this

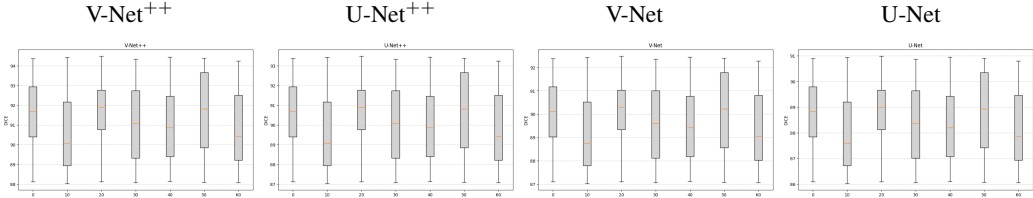

Figure 4. Variation of the dice metric in the MURA-Xray analysis with shifting values of N, which stands for the iterations of the Monte Carlo simulation.

area, which is consistent with the increased values seen on the EUA uncertainty map for the same area. Combining EUA-augmentation with TTD (EUA + TTD) results in the calculation of hybrid uncertainty, which combines impromptu and gnostic uncertainty components.

Table 3. We used VF, URF, and URF$_w$, and measured performance with DICE (%) metrics to evaluate the effectiveness of categorizing musculoskeletal radiographs using various training and testing techniques. At first, we trained without adding further data and got results with several models. We then obtained encouraging results by using data augmentation throughout training. A notable improvement above the baseline of a single prediction is shown by the asterisk (*).

|  |  | VF | URF | URF$_w$ | V-Net$^{++}$ | U-Net$^{++}$ |
|---|---|---|---|---|---|---|
| Training | Baseline | $91.11 \pm 0.97$ | $94.64 \pm 0.49$ | $94.89 \pm 0.59^*$ | $94.16 \pm 0.74^*$ | $94.68 \pm 0.88^*$ |
|  | TTD | $92.36 \pm 0.89$ | $94.87 \pm 0.69$ | $95.16 \pm 0.69^*$ | $94.67 \pm 0.59^*$ | $94.87 \pm 0.56$ |
|  | EUA | $93.67 \pm 0.23$ | $94.89 \pm 0.49$ | $96.59 \pm 0.49^*$ | $95.87 \pm 1.63^*$ | $96.01 \pm 0.23^*$ |
|  | EUA+TDD | $94.36 \pm 0.37$ | $95.69 \pm 0.47$ | $96.84 \pm 0.36^*$ | $94.16 \pm 0.84^*$ | $95.46 \pm 0.36^*$ |
| Training + Aug | EUA | $94.63 \pm 0.52$ | $95.81 \pm 0.53$ | $96.48 \pm 0.41^*$ | $95.23 \pm 0.69^*$ | $95.47 \pm 0.73^*$ |
|  | EUA+TDD | $94.79 \pm 0.29$ | $95.08 \pm 0.79$ | $96.86 \pm 0.28$ | $95.67 \pm 0.83$ | $95.46 \pm 0.89$ |

We used the Dice score and Average Symmetric Surface Distance (ASSD) metrics to conduct a thorough quantitative evaluation of our segmentation results. Four different network architectures – , U-Net Ronneberger et al. (2015), and V-Net Milletari et al. (2016), U-Net$^{++}$ Zhou et al. (2018) and the unique V-Net$^{++}$ are used in conjunction with a variety of testing techniques to conduct these evaluations. We used data augmentation techniques to increase the training dataset for each of these Convolutional Neural Networks (CNNs) during the training phase. The results from the baseline testing approach, which does not use Monte Carlo simulation, were then contrasted with those from TTD, EUA, and EUA + TTD during the inference phase. The first goal of our research was to comprehend how the classification accuracy changes when the number of $N$ Monte Carlo simulation runs is increased.

Our findings showed as depicted in Figure 4 that the segmentation accuracy attained with TTD closely approaches that of the single-prediction baseline across all four network designs. As we increased the value of $N$ from 1 to 10, we saw an improvement in segmentation accuracy for EUA and EUA + TTD. The segmentation accuracy for these two approaches, however, seemed to settle and hit a plateau as $N$ surpassed 20.

## B.1 Quantitative Assessment

We evaluated the performance of TTD and EUA in cases where data augmentation was solely ignored during training, in addition to the prior scenario where augmentation was applied throughout both the learning and testing phases. The quantitative assessments (with $N = 20$) are shown in 4 and take into account different combinations of training and testing techniques. Notably, it becomes clear that EUA constantly has a greater skill in improving segmentation accuracy compared to TTD alone, regardless of whether training entailed data augmentation or not. It is important to note that while combining EUA and TTD does result in increased segmentation accuracy (with a p-value $\geq$ 0.05), this combination is not significantly better than EUA alone.

Table 4. To analyze the effectiveness of musculoskeletal radiograph classification utilizing various training and testing methods, DICE (%) assessments were performed. Initially, we train without data augmentation and acquire the results using different models. Subsequently, we apply data augmentation with training and achieve plausible results. The asterisk (*) denotes a significant improvement over the baseline of a single prediction.

|  | Test | U-Net | V-Net | U-Net$^{++}$ | V-Net$^{++}$ |
|---|---|---|---|---|---|
| Training | **Baseline** | $93.21 \pm 2.12$ | $93.45 \pm 1.98$ | $93.67 \pm 2.06^*$ | $94.02 \pm 1.87^*$ |
|  | **TTD** | $92.98 \pm 2.18$ | $93.15 \pm 2.23$ | $93.80 \pm 2.05^*$ | $94.12 \pm 1.91^*$ |
|  | **EUA** | $92.76 \pm 2.09$ | $93.00 \pm 2.12$ | $93.56 \pm 2.14^*$ | $93.82 \pm 2.01^*$ |
|  | **EUA+TDD** | $93.10 \pm 2.05$ | $93.25 \pm 2.11$ | $93.70 \pm 2.02^*$ | $94.05 \pm 1.95^*$ |
| Training + Aug | **EUA** | $92.84 \pm 2.14$ | $93.12 \pm 2.07$ | $93.65 \pm 2.03^*$ | $94.00 \pm 1.88^*$ |
|  | **EUA+TDD** | $92.95 \pm 2.10$ | $93.18 \pm 2.09$ | $93.75 \pm 2.01^*$ | $94.10 \pm 1.92^*$ |

Table 5. We carried out ASSD assessments to examine the effectiveness of musculoskeletal radiograph categorization using various training and testing methods. In the beginning, we trained models without using data augmentation and got outcomes. We then obtained encouraging results by using data augmentation during training. The asterisk (*) denotes improvements worth mentioning above the baseline of single forecasts.

|  |  | U-Net | V-Net | U-Net$^{++}$ | V-Net$^{++}$ |
|---|---|---|---|---|---|
| Training | **Baseline** | $3.17 \pm 0.32$ | $3.67 \pm 0.29$ | $3.11 \pm 0.25^*$ | $2.87 \pm 0.27^*$ |
|  | **TTD** | $3.07 \pm 0.28$ | $2.97 \pm 0.28$ | $2.84 \pm 0.26$ | $2.15 \pm 0.25$ |
|  | **EUA** | $3.47 \pm 0.36$ | $3.61 \pm 0.32$ | $3.08 \pm 0.34^*$ | $2.11 \pm 0.31^*$ |
|  | **EUA+TDD** | $2.16 \pm 0.23$ | $2.41 \pm 0.22$ | $2.34 \pm 0.19$ | $1.51 \pm 0.20^*$ |
| Training + Aug | **EUA** | $3.15 \pm 0.31$ | $2.78 \pm 0.29$ | $2.38 \pm 0.26^*$ | $2.01 \pm 0.24^*$ |
|  | **EUA+TDD** | $2.13 \pm 0.27$ | $2.06 \pm 0.26$ | $1.56 \pm 0.22^*$ | $1.32 \pm 0.23^*$ |

The Dice score distributions for different representative stacks of musculoskeletal radiographs are shown in this figure 5. The identical U-Net model that was trained utilizing enhanced data was used to get these findings. Notably, for TTD, EUA, and EUA + TTD, Monte Carlo simulations were carried out with a total of 60 runs, in contrast to the baseline approach's single prediction per image. It is clear that the Dice scores acquired using TTD for each example are tightly grouped around the baseline scores. The Dice score distribution for EUA, in comparison, shows a higher average, highlighting EUA's success in enhancing segmentation accuracy. Additionally, the results from EUA are more variable than those from TTD, suggesting that EUA offers more thorough structure-wise uncertainty information. Additionally, Figure 5 shows how closely EUA + TTD's performance matches EUA's.

We performed an evaluation at both the pixel-level and the structure-level to see how our approaches for evaluating uncertainty may assist in identifying improper segmentations. We looked at the combined distribution of pixel-level uncertainty and segmentation errors for TTD, EUA, and EUA + TTD, respectively, in our pixel-level evaluation. By mathematically examining the loss rates of pixels across several pixel-level uncertainty levels within each image slice, this joint histogram was

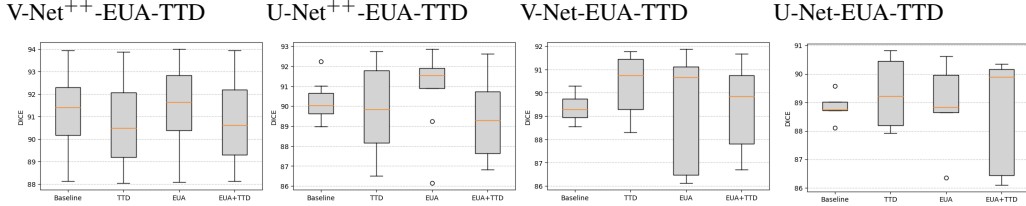

Figure 5. We are examining the Dice distributions of classification outcomes using different testing approaches for various sets of sample image stacks from MURA-Xray.

created. Figure 4 displays the outcomes from the U-Net analysis with a $N$ of 20. To make viewing easier, these joint boxplot have been normalized based on the total number of pixels in the test images.

We measured the average error rate corresponding to various degrees of pixel-wise uncertainty for each category of pixel-level uncertainty, resulting in loss rate curves as a function of pixel-level uncertainty. These curves are shown in Figure 6 as the red curves. The chart shows that most pixels have minimal uncertainty, which is related to a low error rate. The error rate steadily rises as the degree of uncertainty rises. The uncertainty based on TTD (gnostic) is shown in Figure 6. Notably, the error rate increases dramatically when the forecast uncertainty is low. Figure 6 depicts EUA-based uncertainty (impromptu), in contrast, where the rise in error rate happens more gradually. This finding shows that EUA, as compared to TTD, results in fewer instances of overconfident wrong predictions. For various testing methodologies, the dashed ellipses in Figure 4 also show differing degrees of overconfident wrong predictions.

We used Volume Variation Coefficient (VVC) to express structural uncertainty and 1 - Dice score to reflect structural segmentation error for the structural assessment. We show the combined distribution of $\mathbb{V}$ and 1 - Dice score for various validation techniques in Figure 5. U-Net, learned using data augmentation, and N = 19 for reasoning were used for these evaluations. Figure 5 show the findings achieved using TTD, EUA, and EUA + TTD. It's interesting to observe that, for all testing techniques, the $\mathbb{V}$ value tends to grow as the 1 - Dice score rises. The formation shown in Figure 5) is, however, steeper than that seen in Figures 5 (b) and 5 (c). This comparison shows that segmentation error and structural uncertainty estimation supplied by EUA are closely connected, and that EUA causes a greater fluctuation in $\mathbb{V}$ than TTD. EUA and TTD together provide outcomes comparable to those of EUA alone.

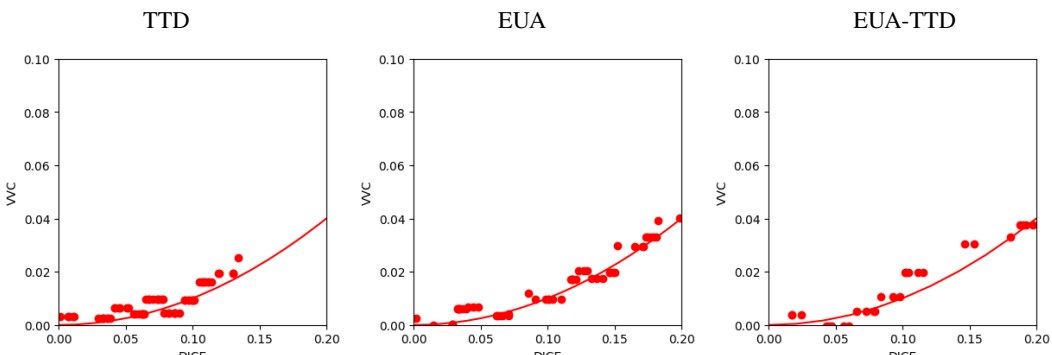

Figure 6. By comparing the $\mathbb{V}$ to 1-Dice coefficient across multiple validation techniques, we are investigating structural uncertainty in the context of MURA classification.

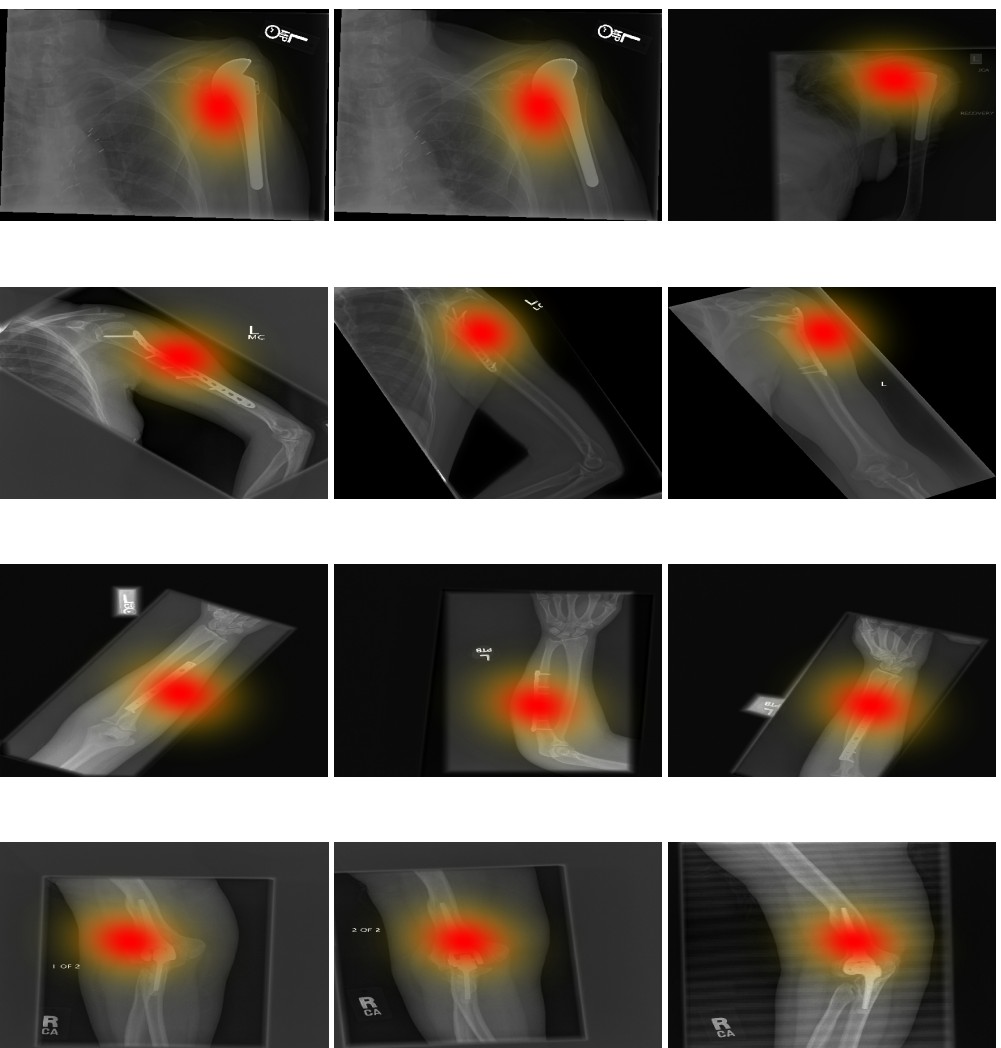

Figure 7. We demonstrate the visualization results of the application of our suggested approach URF$_w$ with EUA-TTD to several classes in the MURA dataset.

