# OpenReview forum: "Enhancing Machine Learning System Reliability in Healthcare through Uncertainty Estimation and Multi-Modal Learning"
_ICLR.cc/2024/Conference — Submitted to ICLR 2024_

### Official Review · Reviewer_8rRf · 2023-10-29

**Soundness:** 2 fair
**Presentation:** 2 fair
**Contribution:** 2 fair
**Rating:** 3
**Confidence:** 3

**Summary:**

This paper delves into an array of uncertainty quantification techniques with the aim of enhancing the amalgamation of predictions from multiple base learners. The authors critically examine the influence of noise and spatial transformations on image-centric tasks and introduce a test-time augmentation (TTA) methodology tailored for uncertainty. Consequently, predictions stemming from this approach manifest heightened accuracy. The proposed entropy-based uncertainty estimate (EUA), when synergized with TTA, outperforms established benchmarks, setting a new baseline in the domain.

**Strengths:**

1. This paper investigates the critical challenge of quantifying uncertainty in medical image segmentation powered by CNNs. The authors undertake a comprehensive exploration of potential solutions, ranging from uncertainty-sensitive fusion strategies to entropy-based uncertainty evaluation techniques. The conceptual foundation is robust, and the approach is astutely anchored in established prior works.

2. The introduced test-time augmentation, when amalgamated with EUA, consistently outshines several benchmarks. This accomplishment not only sets a commendable precedent but also furnishes valuable insights that can pave the way for subsequent research.

**Weaknesses:**

1. It would be helpful to clarify what “modality” refers to in this paper. Most recent studies in the medical domain consider clinical notes, vital signs, tabular data, and medical images as multimodalities. However, this paper focuses solely on CNN-driven medical image segmentation.

 2. The authors should discuss prior works on test time augmentation more thoroughly, for instance, the study found at https://www.nature.com/articles/s41598-020-61808-3. It’s also essential to highlight how the proposed TTA method improves upon previous approaches.

 3. The structure of this paper could be improved. For example, readers must refer to the appendix to access the experiment section, and there are numerous typos. The overall layout does not adhere closely to high-quality writing standards. Therefore, I recommend that the authors significantly revise the current version before submission.

**Questions:**

Please see the strengths and weaknesses parts above.

---

### Official Review · Reviewer_fEUv · 2023-10-30

**Soundness:** 1 poor
**Presentation:** 1 poor
**Contribution:** 1 poor
**Rating:** 1
**Confidence:** 4

**Summary:**

This paper proposes an end-to-end system for the classification of musculoskeletal radiograph images. The system is formulated in the context of multi-modal learning, and comprises an ensemble of methods that are combined by taking into account the uncertainty estimates that they produce.

**Strengths:**

To name one, the broad topic that this paper covers, i.e. uncertainty estimation in medical imaging, is an important one.

**Weaknesses:**

* The whole writing in the paper is very vague and superficial. Just reading the abstract+intro is a clear example of this. Reading the abstract, it seems like the main contributions are 1) the URF technique, 2) an image acquisition model (which seems a bit disconnected from the URF thing, but one expects to find the connection in the intro), 3) the GUE technique. However, nothing is said about the last two points in the intro, completely missing context for the contribution in this sense. Indeed, the first four paragraphs in the intro are very general sentences that do not provide any concrete idea. They just revolve around the idea of "uncertainty is important in medical imaging". Also, the focus in the title and abstract is in "healthcare" (very general); however, in the intro the focus turns way more specific and only "musculoskeletal problems" are mentioned.

* No experiments are provided in the main text. As indicated in the next point, the theoretical/methodological contribution is very minor. Therefore, the interest of the paper would be in an extensive empirical validation of the proposed idea. However, there are no experiments in the main text. There are experiments in the appendix, but pretty disconnected from the main text and lacking a clear experimental setting.

* The methodological contribution, URF, seems to be described in the fourth paragraph of Section 2.1. It is not easy to understand, because the formulation of the problem is again very vague and poorly described (first paragraph of section 2.1). It is also difficult to evaluate the novelty of this idea, because no related work is discussed in detail. They just construct a baseline fusion procedure called Vanilla Fusion (paragraphs 2 and 3 of section 2.1), and propose their URF building on this VF.

* According to section 2.1, second paragraph, "Figure 1 shows a graphic illustration of the VF, URF, and URFw methods". However, after reading the descriptions of these three methods in section 2.1, I still do not see how Figure 1 covers those three approaches. Indeed, nothing about these methods is said in the caption of the figure or in the figure itself.

* Some sentences lack coherence. As a clear example, see section 2.1, third paragraph, first sentence:

"In Vanilla Fusion (VF), the weighting of the loss function is adjusted during training while iteratively boosting among the base learners using loss data, such as Mean Squared Error (MSE) for regression, Kullback-Leibler (KL) divergence loss the discrepancy between the actual distribution of class probabilities and the expected class probabilities and IoU or Jaccard Loss evaluates the overlap between
anticipated and ground truth masks and is frequently used in semantic segmentation problems."

**Questions:**

For future submissions, I would recommend the authors to be more specific about their contributions, clearly define the setting they are tackling, discuss the related work (how other people have addressed the problem before), and leave more room for the experimental validation in the main text.

---

### Official Review · Reviewer_fQud · 2023-11-07

**Soundness:** 1 poor
**Presentation:** 1 poor
**Contribution:** 1 poor
**Rating:** 1
**Confidence:** 4

**Summary:**

The authors state to present techniques that enhance the reliability of machine learning systems in health care. The stated contributions include a technique for uncertainty-aware boosting of ensemble members ("URF"), and extracting uncertainties by computing entropy scores based on test-time Monte Carlo Dropout ("GUE"). Experiments are performed on the MURA data set featuring musculoskeletal radiographs. Based on the employed evaluation metrics, it seems URF is meant to improve calibration and regression performance, while GUE is meant to improve the Dice score of a segmentation task. In the presented evaluation, both methods show improvements against the provided baselines.

**Strengths:**

Addressing the reliability of segmentation systems in practice is a timely topic.

**Weaknesses:**

**Technical contributions vs. related work**


GUE / TTD+EUA:
- The method of running Monte Carlo Dropout at test time and computing entropy scores on the samples for uncertainty estimation in semantic segmentation is presented as a contribution of this work and referred to by the authors first as GUE and later as TTD+EUA. Examples of framing this technique as a contribution can be found in the abstract “We also present Gnostic Uncertainty Estimation (GUE)”, or in the conclusion “The creation of a test-time augmentation-based technique for calculating impromptu uncertainty in medical images that takes into consideration the effects of both image noise and spatial changes was a significant contribution. In addition, we presented a thorough theoretical and mathematical foundation for test-time extension. This methodology included modeling the previous distributions of parameters inside an image acquisition model and creating a prediction distribution using Monte Carlo simulation.”
- However, this technique is widely adopted in the community [1,2,3,4,5,6,7]. What is more irritating is the fact that the sections presenting this technique 2.2 / 2.2.1 / 2.2.2 / 2.2.3 are extremely similar to sections 3.1-3.5 in [1] without adequate reference.

URF:
- Similarly irritating, much of the text in Section 2.1 describing the URF technique is very similar to Section III in [8] (again no reference). It is unclear in what way the proposed URF is different from the uncertainty-aware ensemble boosting in [8]
- Further, it is unclear what the actual purpose of this method is (see paragraph about the missing task definition below). According to the evaluation in the Appendix, it is meant to improve calibration (as measured by BS/ECE/MCE) and regression (as measured by RMSE/MPIW/PICP).
- The calibration task comes as a surprise, the term “calibration” is not mentioned a single time in the main paper. There is no reasoning given for why URF should improve calibration, nor reasoning for why one should care about pixel-wise calibration in segmentation systems in the first place.
- Further, providing evidence that URF improves pixel-wise calibration would require comparing it against actual calibration methods. This is not the case. The compared methods in Table 1 (Pappagari / Sarawgi / Rohanian) are never mentioned outside this Table. When making the effort and checking the literature, I found that none of these methods is proposed or tested for calibration. Actual calibration baselines like a simple temperature scaling are missing.
- The validated regression task in Tables 1 and 2 (as measured by RMSE/MPIW/PICP) also comes as a surprise. “Regression” is mentioned a couple of times in the script as a possible application of the presented methods, but no actual regression task is ever described. Instead, the addressed use-case based on the MURA data set is described as classification: “We provide an end-to-end system specifically created for the classification of fractures musculoskeletal radiographs images, offering a novel and creative method ”
- Overall, there is really not much information to be found about the goals and reasoning behind URF, as the rest of the paper seems to be talking about the unrelated techniques TTD/EUA/GUE/VVC (see e.g. conclusion).

Segmentation backbones:
- Much of the evaluation in the Appendix is spent on looking for improvements in Dice score stemming from either TTD/EUA or different segmentation backbones like the V-Net. Such Dice score improvements are only convincing if they are compared against, or based on, the state-of-the-art medical segmentation method, nnU-Net [9].


**Unacceptable presentation**

- Essential sections are either missing entirely (experimental setup, related work, task definition) or moved to the Appendix (all results figures and tables). The main paper is not readable without the Appendix, forcing reviewers to go through the Appendix in detail. This is an unfair advantage over other ICLR submissions that stick to the page limit.


- The extent of confusion that this paper creates during reading is remarkable and gives the impression that the text has been stitched together from various other places (see also the strong similarity of the method sections to the method sections of related work described above). To give one concrete example of the encountered inconsistencies: Despite thoroughly going through the paper and appendix multiple times, I still do not know which task the authors are trying to solve. Mentions of tasks in the text are a wild mix of categorization, classification, regression, segmentation, and finally, calibration. Thereby, segmentation is not mentioned before Section 2.4, and calibration is only mentioned in the Appendix. Since there is no task definition or experiment description, it is further unclear what the potential targets of categorization or regression could be and which of the tasks is addressed at either the pixel or image level.

- I think anonymity is breached several times throughout the script, revealing the authors’ identities. For instance when talking about "our MURA data set" or the affiliation of the employed radiologists.


- Other examples of inconsistencies and loose ends in the text are:

-- Appendix A elaborately introduces Cohen’s kappa metric, but this metric is never used later.

-- A “Grand Challenge 4” is mentioned in the Summary without context or reference.

-- “Fluky Model” appears in the heading of 2.2.1 but is never mentioned again.

-- The proposed method URF_w is never introduced or defined.

-- Calibration baselines in Table 1 are never introduced or described (Pappagari / Sarawgi / Rohanian).

-- The technique of computing entropy scores based on Monte Carlo dropout is first called GUE and later TTD+EUA.

-- Section 2.2 seems to be about test time augmentations (as it is in [1]), but this is never mentioned.

-- Figure 1 is stated to illustrate the methods VF/URF/URF_w, but it does not.

-- The described problem/motivation, which aims to make this work health-specific (“the occurrence of batch effects”), is never explicitly addressed in the presented work.


**A too-specific application for ICLR**

- There is a discrepancy between, on the one hand, making general methodological claims about enhancing the reliability of classification/regression/segmentation models, and, on the other hand, only studying one specific task: 2D segmentation of musculoskeletal radiographs. It has been widely shown that the diversity of medical data sets requires evaluating such claims on a large number of data sets [9].
- Thus, even if all issues like technical novelty, presentation, and evaluation could be fixed, in my opinion, this work would not be suitable for publication at ICLR.


**References**

[1] https://www.sciencedirect.com/science/article/pii/S0925231219301961

[2] https://arxiv.org/abs/1703.04977

[3] https://www.sciencedirect.com/science/article/pii/S0925231219301961

[4] https://arxiv.org/pdf/2103.16265.pdf

[5] https://arxiv.org/pdf/1811.12709.pdf

[6] https://www.frontiersin.org/articles/10.3389/fnins.2020.00282/full

[7] https://arxiv.org/pdf/2005.14262.pdf

[8] https://arxiv.org/pdf/2104.10715.pdf

[9] https://www.nature.com/articles/s41592-020-01008-z

**Questions:**

Table 3: How do you use Entropy for actual segmentation predictions? How to get a Dice score out of an Entropy score?

Figure 1: How do you use ECE/MCE as a loss function during training?

Appendix: “Based on the test sample results of the distinct modalities, we decide in what order to enhance these models to propagate uncertainty”. What do you mean by this? Basing methodological decisions on test sample results does not sound like a good idea.

---

### Meta-Review · Area_Chair_3XSk · 2023-12-07

**Metareview:**

The reviewers were not convinced by this submission. In particular, they found the experimental results weak compared to the baselines; they found very limited theoretical novelty compared to Wang et al "Aleatoric uncertainty estimation with test-time augmentation for medical image segmentation with convolutional neural networks"; and they raised concerns on the poor quality of the manuscript.

**Justification For Why Not Higher Score:**

Poor quality of writing, unclear theoretical contribution, and weak experimental results.

**Justification For Why Not Lower Score:**

N/A

---

### Decision · Program_Chairs · 2024-01-16

Reject